🔓 | **Open Peer Review** | Epidemiology | Research Article

# Key impact of Beijing strains including new resistant clusters on spread of multidrug-resistant tuberculosis in northern Russia

Yulia Popova,[1] Anna Vyazovaya,[2] Platon Eliseev,[1,3] Elena Miteneva,[1] Dmitrii Polev,[2] Igor Mokrousov,[2] Andrei Mariandyshev[1,4]

**ABSTRACT** Northern Russia is characterized by socio-environmental conditions contributing to the spread of tuberculosis (TB) and a high ~30% rate of primary multidrug-resistant (MDR) TB. We applied high-resolution molecular methods to study the *Mycobacterium tuberculosis* population in the Arkhangelsk region of Northern Russia. All available *M. tuberculosis* isolates recovered from newly diagnosed patients from January to December 2018 ($n = 88$) were genotyped using 24-loci MIRU-VNTR, spoligotyping, and, partly, by whole-genome sequencing (WGS). The population structure revealed a predominance of the Beijing genotype and Euro-American lineage, with significant drug resistance burden associated with Beijing and its B0/W148 strain. Beijing strains showed a significantly higher association with MDR and pre-extensively drug-resistant (pre-XDR) TB compared to non-Beijing strains ($P = 0.0013$). All Beijing B0/W148 isolates were MDR, whereas the majority (71.4%) of Beijing Central Asian/Russian subtype strains were drug-sensitive. WGS analysis of newly discovered Beijing clusters 3828-32 and 10167-32 in this area indicated a historical transmission over several decades, reflecting long-term endemic circulation. The presence of compensatory mutations in *rpoC* among MDR strains suggests enhanced fitness facilitating their ongoing transmission. An intriguing cluster of recent transmission of a non-Beijing strain (spoligotype SIT53, L4.8 sublineage) was identified through combined epidemiological and genomic investigation. To conclude, the prevalence of Beijing strains rose from 39.3% in 1998 to 67.0% ($P < 0.001$), and Russian epidemic MDR strain B0/W148 increased its rate from 11.2% in 1998 to 20.5% ($P = 0.097$). This highlights the key role of MDR Beijing strains, including new resistant clusters, in disseminating MDR-TB in the region and the importance of continuous surveillance using high-resolution genotyping.

**IMPORTANCE** The Arkhangelsk region is the largest province of northern European Russia. One-third of newly diagnosed tuberculosis patients are infected with multidrug-resistant (MDR) *Mycobacterium tuberculosis* strains. We assessed the molecular population structure of *M. tuberculosis* in the Arkhangelsk region in the COVID-19 pre-pandemic year 2018. We identified important MDR clusters and elucidated tuberculosis transmission patterns. An intriguing cluster of recent transmission was identified through the combined use of epidemiological investigation and whole-genome sequencing. The prevalence of Beijing genotype strains increased from 39.3% in 1998 to 67.0%, and the Russian epidemic MDR strain B0/W148 doubled from 11.2% in 1998 to 20.5%. Furthermore, we described new MDR clusters emerging within the Beijing genotype. This highlights the key impact of the MDR Beijing strains and the importance of continuous surveillance using high-resolution genotyping. This study of the pre-pandemic strain collection provides an indispensable intermediate time point between earlier studies carried out 25 years ago and ongoing surveillance.

**KEYWORDS** MIRU-VNTR, spoligotyping, Beijing genotype, drug resistance, genetic diversity, *Mycobacterium tuberculosis*

**Peer Reviewer** Imane Chaoui, National Center of Energy, Sciences and Nuclear Techniques, Rabat, Morocco

Address correspondence to Anna Vyazovaya, annavyazovaya@gmail.com, or Platon Eliseev, pediatrics@yandex.ru.

Igor Mokrousov and Andrei Mariandyshev contributed equally to this article.

The authors declare no conflict of interest.

See the funding table on p. 13.

Tuberculosis (TB), caused by *Mycobacterium tuberculosis*, remains the leading cause of death among infectious diseases and is a major driver of deaths due to antimicrobial resistance (Global tuberculosis report 2024). Globally, an estimated 390,000 people developed multidrug-resistant or rifampicin-resistant TB (MDR/RR-TB) in 2024 (1). The emergence and ongoing spread of drug-resistant strains requires the application of appropriate methods for their identification, tracing, and characterization of drug resistance properties. High-resolution MIRU-VNTR genotyping, single nucleotide polymorphism (SNP) typing, and whole-genome sequencing (WGS) are essential for outbreak investigation, distinguishing relapse from reinfection, and detecting mixed-strain infections (2–4). The 24-locus MIRU-VNTR remains the international standard, whereas spoligotyping serves as an auxiliary tool (5).

Regarding drug resistance detection, WGS is increasingly used as the method of choice for comprehensive drug resistance profiling. The WHO's 2021 catalog of *M. tuberculosis* mutations provides enhanced genotypic resistance detection (6). In July 2023, WHO issued a guidance endorsing targeted next-generation sequencing for drug susceptibility testing (DST) of rifampicin, pyrazinamide, and ethambutol in conjunction with phenotypic testing (7). Low-burden countries such as the UK, the Netherlands, and New York State (USA) have transitioned to WGS for DST, and the US Centers for Disease Control and Prevention sequences every culture-confirmed case. In contrast, WGS adoption in high-burden settings is hindered by economic, logistical, and bioinformatics constraints. Nonetheless, several high-burden countries (Azerbaijan, Bangladesh, Belarus, Pakistan, Philippines, South Africa) are implementing WGS for drug-resistant TB surveillance (8, 9).

*M. tuberculosis* complex comprises 10 human-adapted lineages (L1–L10), nine animal-adapted ecotypes, and one extinct lineage (10). Human *M. tuberculosis sensu stricto* comprises four major lineages (L1–L4). In Russia, lineages L2 (East Asian) and L4 (Euro-American) predominate, while lineages L1 (Indo-Oceanic) and L3 (CAS) are rare (11–16). Beijing genotype is a main part of L2 and is divided into ancient and modern sublineages (17). Both are found in Russia, but the modern Beijing subtypes are predominant and mostly represented by Central Asian/Russian and B0/W148 clusters. The latter was termed a Russian successful clone (18) and is strongly linked to MDR due to compensatory mutations in *rpoA*, *rpoB*, and *rpoC* that mitigate fitness costs of rifampicin resistance mutations such as *rpoB* S450L (19).

Whereas there is no single consensus definition of a medically important cluster, some are considered as such due to their clinical and/or epidemiological significance—that is, association with a more severe disease course, increased virulence (e.g., assessed in animal models), or increased transmissibility (e.g., assessed by high clustering rates or *in vitro* growth rates) (20–22). Indeed, the reasons for success may vary for different clusters/genotypes: some drug-susceptible strains may be important because of increased transmissibility or virulence, as with the LAM SIT264 strain in Russia (23) or one particular susceptible Beijing subtype in South Africa (24). On the other hand, there are hypervirulent but strictly endemic strains with limited spread (e.g., Beijing 14717-15 "Buryat" subtype highly lethal in mouse model [25]). Certain genotypes have unusual manifestations of virulence properties in mouse and macrophage models, such as the mostly drug-susceptible Haarlem genotype (26, 27). Finally, a certain balance between drug resistance and virulence likely resulted in the observed wide dissemination of the Beijing Central Asian/Russian subtype (~94-32 cluster) and the LAM SIT254 spoligotype throughout Russia (23, 25).

The Arkhangelsk region is the largest province by area in northern European Russia. The characteristic features of the region contributing to the increased TB incidence are (i) cold and harsh climate, (ii) high population density in urban areas, and (iii) a non-negligible proportion of the former prison inmates. This region reported significant improvements in TB incidence (from 48.0 to 20.7 per 100,000) and mortality (16.5 → 2.0 per 100,000) from 2000 to 2018 (20, 21). However, primary MDR-TB among new patients rose from 18.7% in 2002 to 33.8% in 2018 (30.4% regionally, 31.9% nationally) (28–30).

The valuable first studies of molecular TB here focused on both community settings and prison populations and demonstrated high clustering rates and particularly active transmission of the Beijing strains in prison (28, 31). These studies showed an increase in Beijing genotype prevalence from 44.5% (1998) to 57.1% (2004–2006), correlating with increasing rate of MDR-TB (28, 32). The recent COVID-19 pandemic has influenced both tuberculosis epidemiology and the population structure of the pathogen (33). Given this context, the assessment of the current *M. tuberculosis* population and its temporal changes requires studying strain collections sampled at multiple time points. In the present study, we used spoligotyping, 24-locus MIRU-VNTR typing, and whole-genome sequencing to assess the genetic diversity of *M. tuberculosis* isolates from the Arkhangelsk region, collected in 2018. We sought to identify MDR-associated clusters and elucidate the transmission patterns of tuberculosis in this northern Russian region in the pre-pandemic period.

## MATERIALS AND METHODS

### Study population

The patients were enrolled within a study of *M. tuberculosis* clinical isolates carried out by the Department of Phthisiopulmonology of the Northern State Medical University and the molecular genetic laboratory of the Arkhangelsk Clinical Anti-Tuberculosis Dispensary. The bacterial isolates were recovered from sputum of patients with pulmonary TB admitted at the Arkhangelsk Clinical Anti-Tuberculosis Dispensary between 1 January and 31 December 2018.

The present study included all newly diagnosed patients with pulmonary TB in 2018 with isolated bacterial culture on solid Löwenstein-Jensen medium (according to Order #951 of 29 December 2014 of the Ministry of Health of the Russian Federation). A total of 198 new TB cases were registered in the general population in the Arkhangelsk region in 2018, of which 184 cases had pulmonary TB. Of them, 151 patients were microscopy and/or culture positive, and bacterial isolate on solid media was available for 90 patients. The classification of clinical forms of tuberculosis, officially endorsed by the Ministry of Health of the Russian Federation, is based on clinical, radiological, and pathomorphological characteristics of tuberculosis, its progression, and bacteriological confirmation (34).

### Drug susceptibility testing

*M. tuberculosis* drug susceptibility testing for isoniazid, rifampicin, ethambutol, pyrazinamide, ofloxacin, kanamycin, capreomycin, cycloserine, and para-aminosalicylic acid was carried out for all strains using the Bactec MGIT 960 system (Becton Dickinson, Sparks, Md.) according to the manufacturer's instructions.

Mutations in the *rpoB*, *katG*, and *inhA* genes associated with resistance to rifampicin and isoniazid were determined using the GenoType MTBDRplus assay (Hain Life Science, Germany). Mutations in the *embB*, *gyrA*, and *rrs* genes associated with resistance to ethambutol, fluoroquinolones, and injectable drugs were detected using Genotype MTBDRsl assay (Hain Life Science, Germany).

The isolate was considered resistant if resistance to the anti-TB drug was detected by at least one of the methods (MTBDRsl or Bactec MGIT). *M. tuberculosis* isolates were considered monoresistant if they were resistant to one of the anti-TB drugs, polyresistant if they were resistant to two or more anti-TB drugs, but not to a combination of isoniazid and rifampicin, and MDR if they were resistant to rifampicin and isoniazid simultaneously, regardless of the presence of resistance to other anti-TB drugs (Order of the Ministry of Health of the Russian Federation No. 951, as amended on 29 October 2014). According to the revised WHO definitions, TB caused by MDR strains resistant to any of the fluoroquinolones was designated as pre-XDR-TB, while pre-XDR strains with additional resistance to bedaquiline or linezolid were designated as XDR (7).

## Genomic DNA extraction and genotyping

*M. tuberculosis* DNA was extracted from bacteria cultured on solid Löwenstein-Jensen medium using the GenoType Mycobacteria Series system and the Cetyltrimethylammonium bromide-based method. Detection of the Beijing genotype, its B0/W148 and Central Asian/Russian, Central Asia Outbreak (CAO) clusters was performed using PCR assays as described (reference 12 and references therein). Strains of other genotypes (non-Beijing) were spoligotyped (35). The spoligoprofiles were compared with the in-house version of the international SITVIT2 database (http://www.pasteur-guade-loupe.fr:8081/SITVIT2/) at the Pasteur Institute of Guadeloupe, and SIT (Spoligotype International Type) was assigned. The spoligotyping profiles of 29 non-Beijing isolates were deposited in the international spoligotype database SITVIT2.

All *M. tuberculosis* strains were subjected to 24-loci MIRU-VNTR typing (36). The digital profiles were compared with the MIRU-VNTRplus database (http://www.miru-vntrplus.org/) and the MIRU-VNTR type was determined according to the MLVA Mtbc 15-9 nomenclature. The same online tool was used for phylogenetic analysis and building the UPGMA dendrogram.

A cluster was defined as two or more isolates sharing an identical spoligotype and/or MIRU-VNTR-24 profile. The Clustering Rate (CR) was calculated by the following formula: $CR = (n_c − c)/n$, where $n_c$ is the total number of clustered strains, $c$ is the number of clusters, and $n$ is the total number of strains (36).

## WGS and bioinformatics analysis

Whole-genome DNA libraries were prepared using the Illumina DNA Prep Kit and sequenced as paired-end reads on the NextSeq 500 platform (Illumina, USA). The resulting *M. tuberculosis* genome sequencing data (FASTQ files) have been deposited in the NCBI Sequence Read Archive under BioProject accession number PRJNA1157023.

SAM-TB online tool (https://samtb.uni-medica.com/index) was used for SNP calling and genotypic detection of drug resistance. On average, 97.86%–98.96% of reads had 10× coverage, with an average depth of 155×–213×.

Geneious 9.0 package (Biomatters Ltd.) was additionally used for mapping the reads to the genome of reference strain H37Rv (NC_000962.3).

## Statistical analysis

A $χ^2$ test was used to detect any significant difference between the two groups. Yates-corrected $χ^2$ and *P*-values were calculated with the 95% confidence interval at http://www.medcalc.org/calc/odds_ratio.php online resource.

## RESULTS

A total of 184 patients with pulmonary tuberculosis were identified in the Arkhangelsk region of Russia from 1 January to 31 December 2018. For 90 of them, *M. tuberculosis* isolates were recovered. Based on further molecular testing, two isolates were excluded from the final analysis. One isolate presented a mixed infection of *M. tuberculosis* and non-tuberculous mycobacteria. For the other isolate, PCR failure was observed under VNTR typing in 13 of 24 VNTR loci. Therefore, we report genotyping data for 88 *M. tuberculosis* isolates from 88 patients.

Fifty-one isolates (57.9%) were susceptible to all tested drugs, and 30 (34.1%) isolates were MDR. Among 30 MDR isolates, the total number of isolates resistant to ethambutol was 23 (79.3%), to injectable drugs (kanamycin, capreomycin) was 15 (51.7%), and to fluoroquinolones was 5 (17.2%). The remaining isolates were INH-monoresistant (*n* = 4), resistant to INH and ethambutol (*n* = 1), and resistant to INH and fluoroquinolones (*n* = 1).

Overall, 59 patients (56.2%) were infected with strains of the Beijing genotype (East Asian lineage, L2). Of 29 non-Beijing strains, 28 belonged to different genotypes of the Euro-American lineage (L4), and 1 isolate belonged to the Central Asian lineage (L3). The

analysis of clinical and epidemiological data of patients, depending on the genotype of *M. tuberculosis* strains, did not reveal statistically significant differences between Beijing and non-Beijing strains (Table 1).

All Beijing strains belonged to the modern sublineage of the Beijing genotype. Furthermore, 18 (20.5%) and 35 (39.8%) isolates were assigned to the Beijing B0/W148 and Beijing Central Asian/Russian subtypes (the latter included two isolates of the CAO clade), respectively (Table 2).

## Drug resistance by genotype

Strains of the Beijing genotype were more frequently associated with multidrug and pre-extensive drug resistance (MDR/pre-XDR) compared to non-Beijing strains (47.5%; 28/59 vs non-Beijing 6.9%; 2/29) (OR = 12.19 [2.65–56.00]; *P* = 0.0013). All B0/W148 subtype isolates were MDR, and three of them were pre-XDR (Table 2). The Central Asian Russian subtype included strains with diverse resistance patterns, and the proportion of MDR isolates was only 17.1% (6/35). Both CAO isolates were drug-resistant, and one of them was pre-XDR.

Non-Beijing genotypes included various families (L4-unclassified/T, LAM, Ural, Haarlem, Cameroon, CAS1-Delhi) and were predominantly drug-sensitive (86.2%).

In all 30 MDR/pre-XDR isolates of *M. tuberculosis*, regardless of genotype (Beijing or non-Beijing), the *katG* Ser315Thr mutation, a molecular marker of isoniazid resistance, was detected. Additionally, in one LAM isolate (SIT803), this mutation co-occurred with *inhA* T(–8)C. Regarding rifampicin resistance, the *rpoB* Ser531Leu mutation was most frequently identified, being present in 89.3% (25/28) of Beijing strains and 100% of non-Beijing strains. Less common among Beijing isolates were the *rpoB* Leu511Pro (subtype Central Asian Russian), *rpoB* His526Asn (clade CAO), and *rpoB* His526Tyr (subtype Central Asian Russian) mutations, each observed in only one case.

## VNTR clustering and drug resistance in Beijing isolates

MIRU-VNTR typing of 59 Beijing isolates identified 24 variants (HGI 0.79), with nine profiles shared by more than one isolate (Fig. 1). The largest cluster, Mlva 94-32, encompassed 23.7% (14/59) of Beijing isolates, including two CAO subtype isolates. Overall, the majority of Central Asian/Russian isolates (71.4%; 25/35) were distributed among six VNTR clusters, yielding a CR of 0.54. Among the B0/W148 isolates, 83.3% (15/18) grouped into two VNTR clusters: 100-32 (44.4%; 8/18) and 3828-32 (38.9%;

**TABLE 1** Demographic and clinical characteristics of patients infected by Beijing and non-Beijing *M. tuberculosis* isolates

| Characteristic | Total | Beijing | Non-Beijing | *P*-value |
| --- | --- | --- | --- | --- |
| | *n* = 88 (%) | *n* = 59 (%) | *n* = 29 (%) | |
| Gender | | | | |
| Male | 64 (72.7) | 42 (71.2) | 22 (75.9) | 0.644 |
| Female | 24 (27.3) | 17 (28.8) | 7 (24.1) | |
| Age groups | | | | |
| 18–30 | 2 (2.3) | 7 (11.7) | 2 (6.9) | 0.890 |
| 31–40 | 13 (14.8) | 16 (26.7) | 7 (24.1) | |
| 41–50 | 31 (35.2) | 18 (30.0) | 11 (37.9) | |
| 51–60 | 21 (23.9) | 13 (22.0) | 7 (24.1) | |
| ≥60 | 3 (3.4) | 6 (10.2) | 2 (6.9) | |
| HIV positive | 8 (9.0) | 6 (10.2) | 2 (6.9) | 0.632 |
| Clinical forms of TB | | | | |
| Infiltrative TB | 58 (65.9) | 38 (64.4) | 20 (69.0) | 0.339 |
| Disseminated TB | 21 (23.9) | 13 (22.0) | 8 (27.6) | |
| Focal TB | 2 (2.3) | 1 (1.7) | 1 (3.4) | |
| Fibrous-cavernous TB | 2 (2.3) | 2 (3.4) | | |
| Other | 5 (5.7) | 5 (8.5) | | |

**TABLE 2** Distribution of drug resistance in *M. tuberculosis* genotypes

| Genotype, subtype, or family | Sensitive, number | Mono-/polydrug resistance, number | MDR/pre-XDR[b] number | Total, number (%) |
|---|---|---|---|---|
| Beijing | 26 | 5 | 28 | 59 (67.0) |
| B0/W148 | 0 | 0 | 18/3 | 18 (20.5) |
| Central Asian Russian[a] | 25 | 4 | 6/2 | 35 (39.8) |
| Central Asia Outbreak | 0 | 1 | 1/1 | 2 (2.3) |
| Other Modern | 1 | 1 | 4 | 6 (6.8) |
| Non-Beijing | 25 | 2 | 2 | 29 (33.0) |
| L4-unclassified/T[c] | 12 | 0 | 0 | 12 (13.6) |
| LAM | 1 | 1 | 2 | 4 (4.5) |
| Ural | 4 | 1 | 0 | 5 (5.7) |
| Haarlem | 6 | 0 | 0 | 6 (6.8) |
| Cameroon | 1 | 0 | 0 | 1 (1.1) |
| CAS1-Delhi | 1 | 0 | 0 | 1 (1.1) |

[a]Central Asian Russian including CAO.
[b]MDR including pre-XDR.
[c]A group designated "L4-unclassified/T" includes strains assigned to ill-defined T family by spoligotyping and/or clustered within the L4 under phylogenetic analysis with MIRU-VNTRplus.org reference profiles.

7/18) (CR 0.72). Additionally, six isolates (10.2%; 6/59) belonging to the modern Beijing sublineage but unrelated to the Central Asian/Russian and B0/W148 genotypes included four isolates in the 10167-32 cluster (CR 0.50).

Susceptibility to all tested drugs was observed in 78.6% (11/14) of isolates of the Central Asian/Russian cluster 94-32 and in all isolates of clusters 11427-32, 7188-32, and 9397-32. All 10167-32 cluster isolates were MDR (Fig. 1). All isolates in the 3828-32 cluster and 75% of isolates in the 100-32 cluster exhibited resistance to ethambutol. In the Central Asian/Russian sublineage, only 6/35 (17.1%) were MDR, distributed among clusters 94-32, 1065-32, and 213-32. Injectable-drug resistance occurred in 8/18 (44.4%) B0/W148 versus 4/35 (11.4%) Central Asian/Russian (*P* = 0.008).

## WGS analysis of Beijing 3828-32 and 10167-32 clusters

To gain more in-depth genomic insight into transmission and drug resistance characteristics, we performed WGS on the isolates of the Beijing 3828-32 (*n* = 7) and 10167-32 (*n* = 4) clusters. These clusters were selected because (i) all their isolates were MDR, hence of high clinical relevance, and (ii) they have not been previously described in northwestern and northern Russia.

All 11 isolates of these clusters were submitted to WGS, but the fastq data were of low quality for one of the Beijing 3828-32 isolates and thus were not included in the further analysis.

Application of the SNP-barcode system showed that cluster 3828-32 strains belong to L.2.2.M4.5 (= B0/W148 subtype). In turn, cluster 10167-32 strains belong to L.2.2.M2.5, which corresponds to Beijing Asian African 2 (37).

Comparison of genotypic drug resistance profiles obtained using WGS showed that all strains of the 3828-32 cluster have identical mutations in the genes conferring resistance to isoniazid, rifampicin, streptomycin, ethambutol, and PAS (Table 3). Three 3828-32 cluster strains had different mutations conferring resistance to PZA, two of which had mutations in genes associated with resistance to injectable anti-TB drugs.

All strains of the other cluster, 10167-32, had identical mutations, *rpoB* S450L, *katG* S315T, *rpsL* K43R, and *ethA* (frameshift), conferring resistance to INH, RIF, STR, and ethionamide, respectively. In addition, three strains also had mutations in genes determining resistance to injectable drugs, and two strains had PZA resistance mutations.

A search for previously described compensatory mutations revealed *rpoC* Val483Gly in five MDR strains of cluster 3828-32 and *rpoC* Ile491Val in one isolate. All four MDR

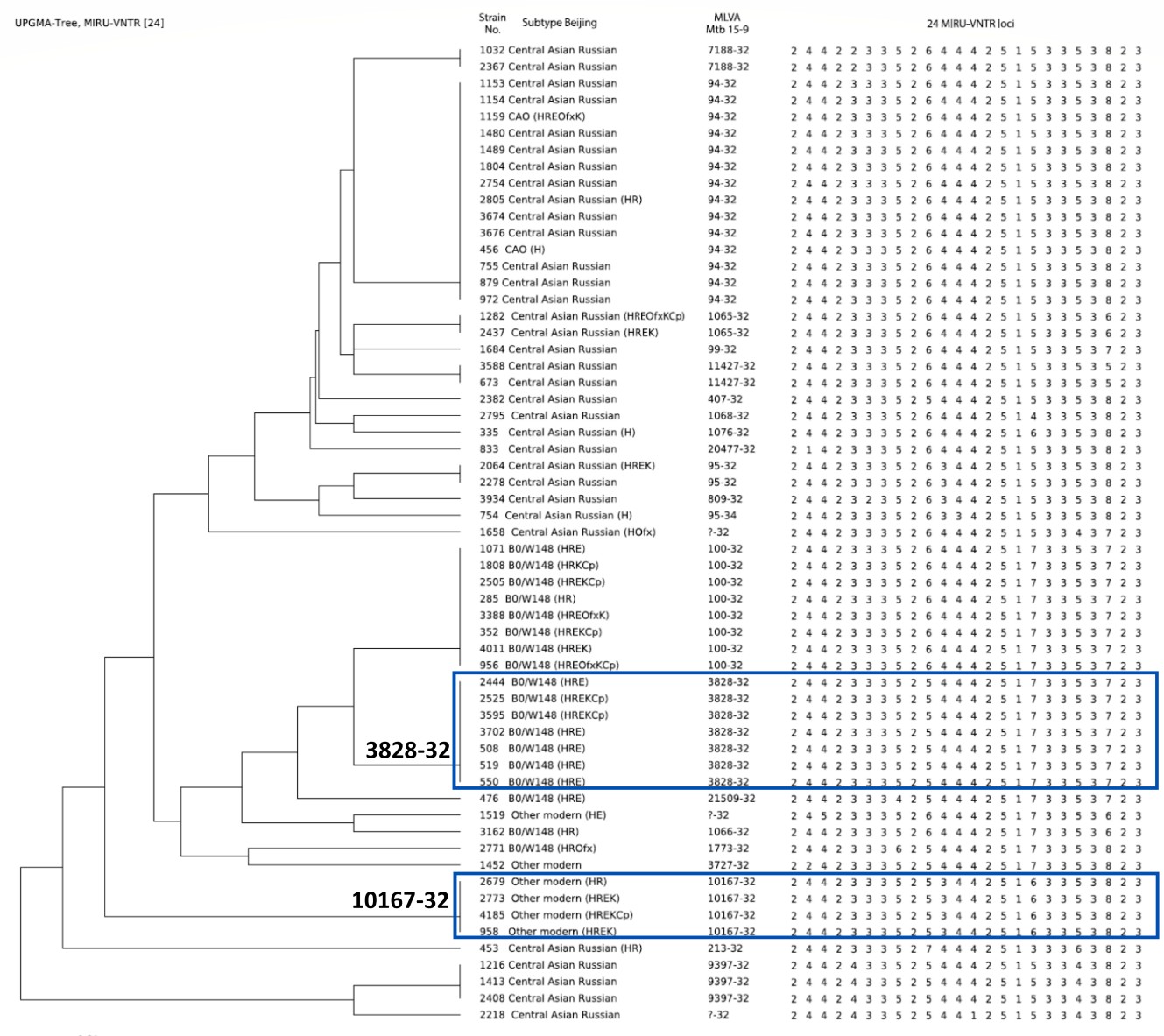

**FIG 1** UPGMA dendrogram of 59 *M. tuberculosis* Beijing genotype isolates based on 24-locus MIRU-VNTR. The Mlva15-9 codes are according to the MIRU-VNTRplus database; the order of loci is clockwise, starting with VNTR154. The drug resistance profile is shown in parentheses. H, isoniazid; R, rifampicin; E, ethambutol; K, kanamycin; Cp, capreomycin.

strains of cluster 10167-32 (Asian African 2 Beijing) had a compensatory mutation, *rpoC* Leu516Pro (Table 3).

Figure 2 presents the pairwise SNP distances among strains within clusters 3828-32 and 10167-32, calculated from WGS data. The isolates were separated by seven or more SNPs, which is above the five SNPs threshold commonly used as a marker of recent transmission (38).

Within cluster 3828-32, SNP differences ranged from 11 to 24, which—assuming an average mutation rate of ~0.5 SNPs per genome per year (38)—translates to approximately 20–50 years since the most recent common ancestor. This suggests that these strains have been circulating locally since the 1970s through the late 1990s. In cluster 10167-32, pairwise distances of 7–16 SNPs correspond to ~15–30 years to the common ancestor (i.e., 1990s to early 2000s), again indicating historical (and ongoing) rather than recent transmission.

**TABLE 3** Major drug resistance and compensatory mutations of *M. tuberculosis* strains

| Strain | MLVA Mtbc 15-9 | Drug resistance mutations | | | | | | | | Compensatory |
|---|---|---|---|---|---|---|---|---|---|---|
| | | RIF (*rpoB*) | INH (*katG*) | STM (*rpsL*) | EMB (*embA*; *embB*) | PZA (*pncA*) | PAS (*folC*) | AMK, CAP, KAN (*rrs, eis*) | ETO (*ethA*) | mutations (*rpoC*) |
| 3595 | 3828-32, L.2.2.M4.5 | S450L | S315T | K43R | *embB* M306V | Q141P | I43T | *eis* G-12A | | V483G |
| 508 | 3828-32, L.2.2.M4.5 | S450L | S315T | K43R | *embB* M306V | | I43T | | | V483G |
| 519 | 3828-32, L.2.2.M4.5 | S450L | S315T | K43R | *embB* M306V | | I43T | | | V483G |
| 550 | 3828-32, L.2.2.M4.5 | S450L | S315T | K43R | *embB* M306V | | I43T | | | I491V |
| 3702 | 3828-32, L.2.2.M4.5 | S450L | S315T | K43R | *embB* M306V | Frameshift | I43T | | | V483G |
| 2525 | 3828-32, L.2.2.M4.5 | S450L | S315T | K43R | *embB* M306V | I133T | I43T | *rrs* A1401G | | V483G |
| 2773 | 10167-32, L.2.2.M2.5 | S450L | S315T | K43R | *embA* C-16T, G406S | T76P | | *eis* C-10T | Frameshift | L516P |
| 958 | 10167-32, L.2.2.M2.5 | S450L | S315T | K43R | *embA* C-16T, G406S | T76P | | *eis* C-10T | Frameshift | L516P |
| 4185 | 10167-32, L.2.2.M2.5 | S450L | S315T | K43R | *embA* C-16T | | | *rrs* A1401G | Frameshift | L516P |
| 2679 | 10167-32, L.2.2.M2.5 | S450L | S315T | K43R | *embA* C-16T | | | | Frameshift | L516P |

## Genotyping of non-Beijing strains

Among 88 *M. tuberculosis* isolates, 29 belonged to non-Beijing lineages (Table 2; Fig. 3). Combined 24-locus MIRU-VNTR and spoligotyping assigned these 29 isolates to Euro-American lineage families: Haarlem (*n* = 6), Ural (*n* = 5), LAM (*n* = 4), and Cameroon (*n* = 1), while 12 isolates with T family spoligotypes were designated as L4-unclassified. One isolate belonged to the CAS1-Delhi genotype of the Central Asian lineage.

SIT53 was the most frequent spoligoprofile among non-Beijing strains (8/29; 27.6%), while SIT52, SIT35, SIT50, SIT46, and a novel profile, SIT4252, each accounted for two isolates. Antimicrobial susceptibility testing revealed that 25/29 (86.2%) non-Beijing isolates were fully susceptible. Two LAM family isolates (both SIT4252 and Mlva 121-52) were MDR with additional ethambutol resistance (Fig. 3).

Overall, 22 distinct MIRU-VNTR profiles and 17 spoligotypes were identified, forming five clusters (Fig. 3). A retrospective epidemiological investigation examined patient data for isolates within each of these clusters. No epidemiological links were found among cases within four clusters.

However, for cluster #2 (Fig. 3), intriguing epidemiological links were identified. This cluster included three isolates (189, 2493, and 3672), all of which belonged to the SIT53 spoligotype and L4.8 sublineage based on WGS data and SNP barcoding. All three patients of this cluster were males and residents in a small town, Nyandoma.

**Beijing B0/W148 3828-32 MLVA Mtbc 15-9  L.2.2.M4.5**

**Beijing 10167-32 MLVA Mtbc 15-9  L.2.2.M2.5**

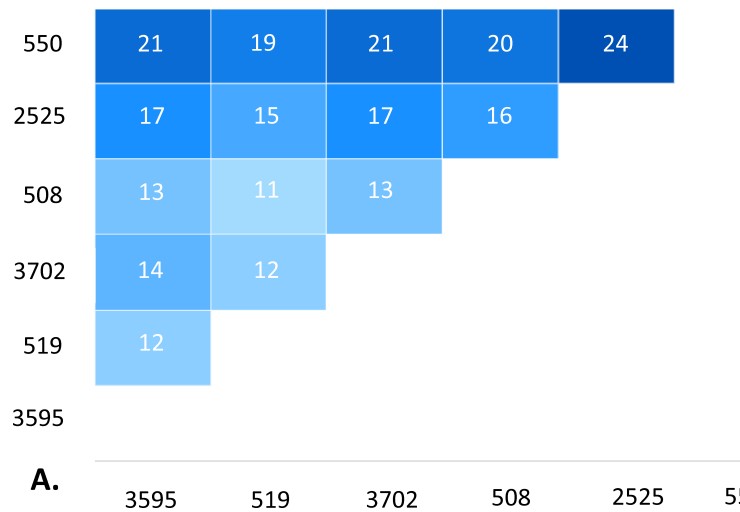

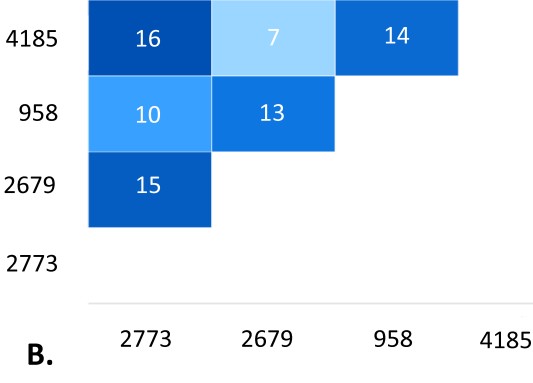

**FIG 2**  Pairwise SNP distances within newly detected MDR VNTR clusters: (A) 3828-32 cluster; (B) 10167-32 cluster.

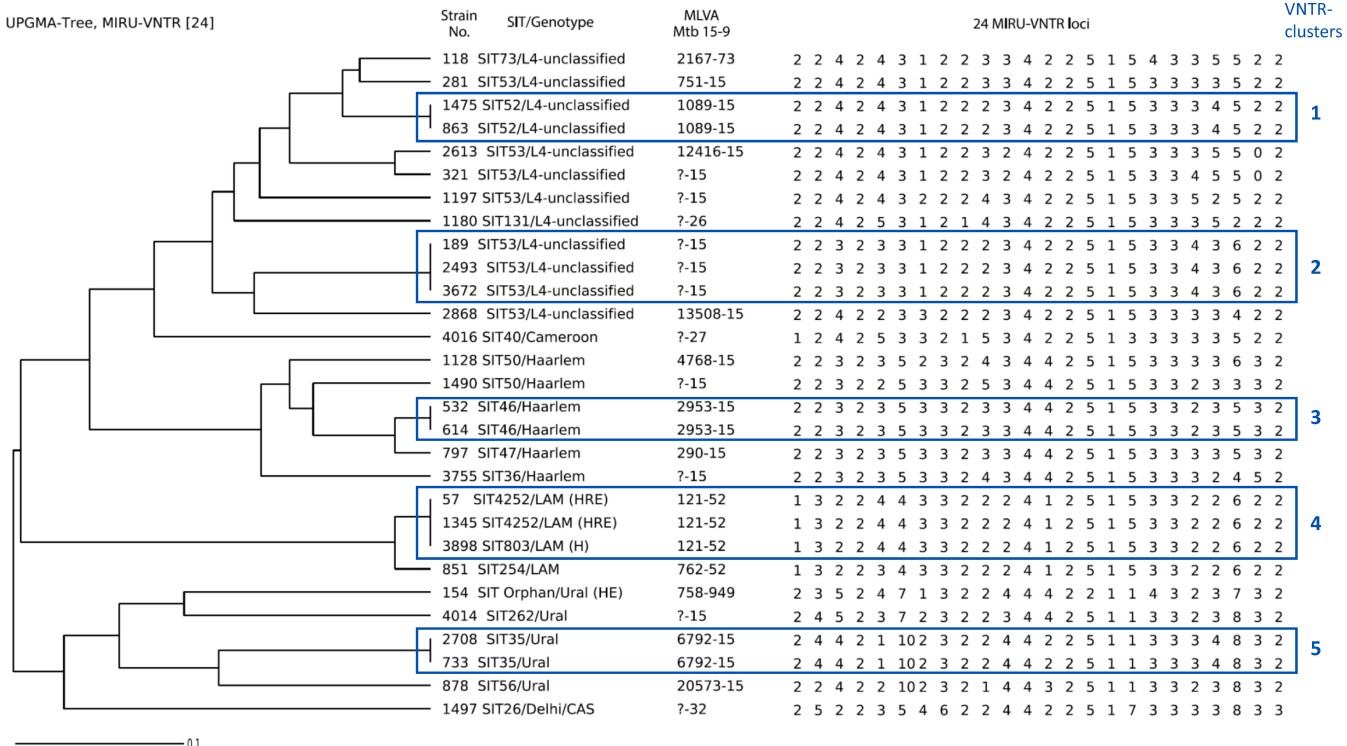

**FIG 3** UPGMA dendrogram of 29 non-Beijing *M. tuberculosis* isolates based on 24-locus MIRU-VNTR (the Mlva15-9 codes are according to the MIRU-VNTRplus database; the order of loci is clockwise, starting with VNTR154).

According to standard epidemiological investigation, they were not directly linked but could be connected through a common source, a hypothetical index case with whom they had a proven contact. That common contact was a 40-year-old female. She had a drug-susceptible TB newly diagnosed in 2015 and reportedly cured in January 2016; unfortunately, no bacterial isolate was available for this study. A more in-depth study revealed that she was likely infected with TB in prison in 2013, with first symptoms manifesting in December 2013, and was released in early 2014 as smear-negative. However, she continued to be followed up in the local TB dispensary and was found smear-positive in July 2014, further diagnosed with TB in 2015 and ultimately cured in 2016. At unknown time points, she had a close household contact with two patients of cluster #2 (isolate 189, 17.01.2018, and 3672, 07.11.2018). In 2017, she had a family contact with the third patient (isolate 2493, 30.07.2018). Based on the WGS data, these

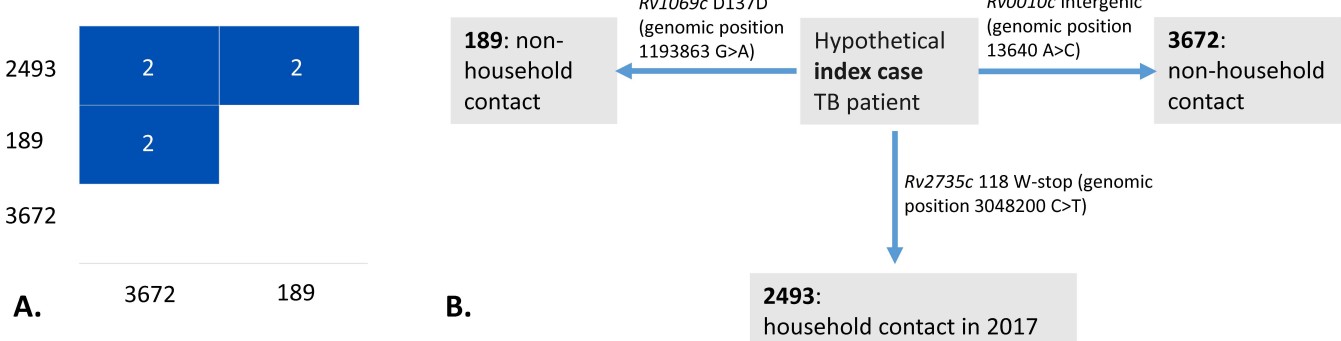

**FIG 4** WGS analysis of three VNTR-clustered non-Beijing strains of VNTR cluster 2 (Fig. 3). (A) Pairwise SNP distances between VNTR-clustered non-Beijing strains; (B) possible links based on WGS data. Hypothetical index case: female, 40 years old, possibly infected with TB in prison in 2013; first symptoms in December 2013; released in early 2014, smear-negative; in July 2014, smear-positive. Newly diagnosed with TB in 2015; cured in 2016.

three isolates are separated by two SNPs in all three pairwise comparisons. In fact, each isolate was separated from a hypothetical ancestral node by one SNP (Fig. 4).

## DISCUSSION

The first population study of *M. tuberculosis* in Arkhangelsk was conducted 25 years ago using major typing methodologies of that time, IS*6110*-RFLP and spoligotyping, and demonstrated the prevalence of the Beijing genotype (28). The present study analyzed *M. tuberculosis* strains collected during 2018. In that year, Arkhangelsk reported the highest MDR-TB rates among newly diagnosed patients compared to all other provinces in Northwestern Russia. We hypothesized that these elevated resistance rates could be attributed to changes in the local population structure of the pathogen. Indeed, the present molecular analysis of *M. tuberculosis* strains isolated from all newly diagnosed tuberculosis patients in this region revealed significant shifts in genotype distribution and resistance patterns, providing insights into the drivers of MDR-TB in this region.

*M. tuberculosis* strains circulating in the Arkhangelsk region are predominantly composed of phylogenetic lineages L2 (Beijing) and L4 (Euro-American), consistent with the genotype distribution observed throughout Russia and reflecting their global prevalence (11, 13–15). At the same time, the proportion of Beijing strains among newly diagnosed patients rose from 39.3% in 1998–1999 (31) to 57.1% among newly diagnosed patients in 2004–2006 (32), reaching 67.0% in our 2018 cohort ($P < 0.001$, for comparison between 1998 and 2018). This upward trend strikingly parallels the concurrent increase in MDR-TB rates among new cases, which increased from 13.5% in 1998–1999 to 28.6% in 2005 and 33.8% in 2018 (28, 30, 32). Given the epidemiological significance of the B0/W148 subtype, we performed a retrospective analysis of IS*6110*-RFLP patterns from the original 1998–1999 study (39) and revealed that B0/W148 strains comprised 16.8% of Beijing isolates (combined previously treated and newly diagnosed patients) compared to 20.5% in our study ($P = 0.296$). Comparison of the data only for newly diagnosed patients showed a double increase in the rate of B0/W148 from 11.2% (10/89) in 1998 to 20.5% in our study ($P = 0.097$). This highlights a central role of this Russian Beijing subtype in disseminating MDR-TB in the region, especially as all B0/W148 isolates were MDR, whereas the majority (71.4%) of Beijing Central Asian/Russian subtype strains were drug-sensitive.

The strains of the non-Beijing genotypes accounted for one-third of the studied collection and demonstrated considerable genetic diversity. They mostly belonged to Euro-American phylogenetic lineage (LAM, Haarlem, Ural), whereas a substantial proportion (13.6% of the entire collection) was defined as L4-unclassified strains. The predominant drug susceptibility among non-Beijing strains, with MDR detected only in two unrelated strains of SIT4252 LAM family, contrasts sharply with the Beijing-associated resistance burden.

An important finding of this study is the identification of two previously undescribed VNTR clusters associated with MDR-TB in Northwestern Russia. The 3828-32 cluster (B0/W148 subtype) and 10167-32 cluster (Beijing Asian African 2 subtype) represent novel epidemiologically important strains and require particular attention. While strains of cluster 3828-32 were previously detected in limited numbers in Primorsky Krai, Far Eastern Russia (40), and sporadic MDR strains were reported in Estonia, Italy, the Netherlands, and Sweden (41), our study represents the first description of their circulation and resistance characteristics in Northwestern Russia.

The VNTR type 3828-32 belongs to the Beijing B0/W148 subtype, also termed the "Russian successful strain" (18). Two B0/W148-specific mutations were previously identified: *whiB6* T51P and ΔTG in the *kdpD* gene (positions 2543–2544). These genes are transcriptional regulators, and these mutations—especially the one in *kdpD*—were speculated to be linked to the virulence of this strain (42). However, a recent study of the H37RvΔ*whiB6* and H37RvΔ*kdpDE* constructs found that neither mutation led to hypervirulence in mice nor impaired bacterial growth *in vitro* (43). We checked the WGS data of the 3828-32 strains from Archangelsk and found that all of them have these

particular *whiB6* and *kdpDE* mutations. In this sense, they do not differ from other Beijing B0/W148 strains. Perhaps a more precise study of the impact of these mutations would require the analysis of gene constructs based not on the laboratory strain H37Rv but on the more closely related Beijing genotype strain—for example, Central Asian/Russian subtype (MIRU-VNTR 94-32 cluster). It might be that the high spread potential of certain epidemic strains is not linked to increased virulence factor expression or secretion, but to other factors related to other aspects of bacterial-human coadaptation (44, 45). A recent transcriptomic and phylogenomic study found that many clinical isolates harbor variants associated with decreased expression of virulence factors EsxA (Esat6) and EsxB (Cfp10), and that these variants were associated with increased transmissibility, especially in drug-resistant strains (44).

Other cluster 10167-32 (four isolates) belongs to the Beijing Asian African 2 subtype (L2.2.M2.5), which is particularly noteworthy, as this subtype has not been previously described in Northwestern Russian regions (12). In Europe, three MDR strains, 10167-32 MtbC15-9, were isolated in Lithuania in 2010–2011 (42). In Russia, only two strains of the Asian African 2 subtype were reported in one study in Eastern Siberia, but their presence could be explained by bordering Mongolia, where this lineage is endemic and prevalent (46). This finding suggests either recent introduction or previously unrecognized circulation of this lineage in Northwestern Russia.

In spite of the emergence of the new cluster 10167-32, "Asian Russia" strains were rare or absent in the studied collection of northern Russia. For example, we did not detect any strain of the ancient Beijing sublineage, whereas two MDR/pre-XDR clusters of such strains were discovered in recent years in Siberia and the Far East (47). On the other hand, CAO and AA2 strains were detected in two and four isolates, respectively, which contrasts with their endemic prevalence in the Siberian Russia's neighbors Kazakhstan and Mongolia, respectively. This supports a hypothesis that a strain actively transmitted in one location does not necessarily become epidemic in a new setting (18, 48).

Our WGS analysis revealed that both new Beijing clusters 10167-32 and 3828-32 represent historical rather than recent transmission events. Based on SNP distances, these highly resistant *M. tuberculosis* strains have been circulating in the region since the 1970s–1990s (cluster 3828-32) and 1990s–early 2000s (cluster 10167-32), contributing to the current MDR tuberculosis burden.

We compared our WGS results on clusters 3828-32 and 10167-32 with an EUSeqMyTB SNP panel developed for identification of cross-border MDR-TB clusters (49, 50). In that study, 56 cross-border clusters (snpCLs) were described in the EUSeqMyTB 2020 pilot project which involved whole-genome sequencing of 2,218 rifampicin-resistant and multidrug-resistant *M. tuberculosis* isolates from 25 EU/EEA countries (49). However, neither 3828-32 nor 10167-32 corresponded to these European clusters.

One of the key determinants of the current MDR-TB epidemic is the ability of resistant strains to maintain fitness comparable to drug-susceptible strains. Although resistance mutations in *rpoB* and *katG* often reduce bacterial viability, compensatory mutations in *rpoC*, *rpoA*, and the *oxyR-ahpC* intergenic region can restore fitness (51, 52). Notably, the *rpoC* Phe452Ser mutation previously identified in MDR Beijing strains of the large MIRU-VNTR cluster ECDC0002 (= B0/W148) serves as a compensatory adaptation for the common *rpoB* Ser450Leu mutation (53). In our MDR isolates, compensatory mutations were confined to *rpoC*. All strains of cluster 10167-32 (L.2.2.M2.5, Beijing Asian African 2) carried the *rpoC* Leu516Pro substitution, which mitigates the fitness cost of *rpoB* Ser450Leu and attenuates its impact on RNA polymerase function (54). In turn, all cluster 3828-32 strains harbored *rpoC* Val483Gly, a mutation that has arisen independently in multiple lineages and likely represents convergent compensation for *rpoB* S450L (55).

## Limitations

This study had certain limitations. First, the observation period was relatively short (1 year). Second, *M. tuberculosis* isolates using solid media were obtained for half of pulmonary cases (90 out of 184), while the remaining cases either did not have

bacteriological confirmation at all or did not show growth on solid media; nonetheless, all available isolates within the survey period were included.

## Conclusion

This study highlights the continued predominance of the Beijing genotype of *M. tuberculosis* in Northern Russia, with alarming levels of drug resistance concentrated within specific sublineages. The proportion of the most medically important Beijing genotype among newly diagnosed patients rose from 39.3% in 1998–1999 (31) to 67.0% in our 2018 cohort ($P < 0.001$). Furthermore, the rate of the epidemic Beijing subtype B0/W148 doubled from 11.2% in 1998 to 20.5% in our study ($P = 0.097$).

The prevalence of MDR among Beijing B0/W148 strains, alongside the presence of compensatory mutations and high clustering rates, suggests ongoing transmission of drug-resistant and adapted strains in the Arkhangelsk region. In addition to the notorious Beijing Mlva 100-32 strains (Russian epidemic clone B0/W148), new endemic MDR clusters were identified. Mlva 3828-32 is a part of B0/W148. In its turn, 10167-32 belongs to the Asian African 2 Beijing clade that is very rare in Russia as a whole and has never been described in northern Russia but is prevalent in Mongolia (6,200 km distance). This raises the question of when and how these strains were brought to this region.

An intriguing cluster of recent transmission due to a non-Beijing strain (spoligotype SIT53, L4.8 sublineage) was identified through the combined use of epidemiological and genomic methods. Three patients were likely linked to a hypothetical index case by both household and non-household contact events, which highlights the diversity of the routes of transmission of the pathogen.

The identification of long-term endemic circulation of highly drug-resistant clusters of Beijing genotype strains, along with their high clustering rate (24-VNTR-based CR 0.72 for B0/W148 vs 0.20 for Central Asian/Russian MDR strains), indicates active transmission of adapted drug-resistant strains. This emphasizes the need to monitor the circulation of the B0/W148 subtype and other identified Beijing clusters associated with MDR-TB.

In the present study, we sought to compare pre-COVID-19 pandemic data (the "old normal") with data published in the early 2000s. In this context, a particular point of interest is that our study provides an important intermediate time point between earlier studies carried out 20–25 years ago and ongoing prospective surveillance. In a continuation study, we plan to include *M. tuberculosis* isolates collected during the pandemic and at least 5 years after it was declared ended in 2023 (the "new normal"). This will provide a comprehensive view of possible spatiotemporal changes in the *M. tuberculosis* molecular population structure in this northern Russian region and the underlying factors that shape it.

## ACKNOWLEDGMENTS

We thank David Couvin for comparison with the updated version of the SITVIT2 database at the Pasteur Institute of Guadeloupe.

We acknowledge support from the Russian Science Foundation (grant 24-44-00004).

## AUTHOR AFFILIATIONS

[1]Northern State Medical University, Arkhangelsk, Russia

[2]St. Petersburg Pasteur Institute, St. Petersburg, Russia

[3]National Medical Research Center for Phthisiopulmonology and Infectious Diseases, Moscow, Russia

[4]Northern Arctic Federal University, Arkhangelsk, Russia

## AUTHOR ORCIDs

Anna Vyazovaya  http://orcid.org/0000-0001-9140-8957
Platon Eliseev  http://orcid.org/0000-0001-9039-4557
Dmitrii Polev  http://orcid.org/0000-0001-9679-2791
Igor Mokrousov  http://orcid.org/0000-0001-5924-0576
Andrei Mariandyshev  http://orcid.org/0000-0002-8485-5625

## FUNDING

| Funder | Grant(s) | Author(s) |
| --- | --- | --- |
| Russian Science Foundation | 24-44-00004 | Anna Vyazovaya |
| | | Igor Mokrousov |

## AUTHOR CONTRIBUTIONS

Yulia Popova, Formal analysis, Investigation, Writing – original draft | Anna Vyazovaya, Conceptualization, Formal analysis, Investigation, Writing – original draft | Platon Eliseev, Conceptualization, Formal analysis, Supervision, Writing – review and editing | Elena Miteneva, Investigation | Dmitrii Polev, Investigation | Igor Mokrousov, Formal analysis, Writing – original draft | Andrei Mariandyshev, Conceptualization, Supervision, Writing – review and editing

## DATA AVAILABILITY

All data contained in this study can be obtained from the corresponding author upon reasonable request. The *M. tuberculosis* whole-genome sequencing data (FASTQ files) were deposited in the NCBI Sequence Read Archive under BioProject accession number PRJNA1157023.

## ETHICS APPROVAL

The study was approved by the Ethics Committee of Northern State Medical University (Arkhangelsk, Russia; Protocols 09/12-13 of 11 December 2013, and 09/10-24 of 22 October 2024). Informed consent was obtained from all subjects and/or their legal guardians. No patient-related data were used in this study. The research was conducted in accordance with the Declaration of Helsinki.

## ADDITIONAL FILES

The following material is available online.

Open Peer Review

**PEER REVIEW HISTORY (review-history.pdf).** An accounting of the reviewer comments and feedback.

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
