## [Reviewer comments · Microbiology Spectrum]

Microbiology Spectrum

Key impact of Beijing strains including new resistant clusters on spread of multidrug-resistant tuberculosis in northern Russia

Yulia Popova, Anna Vyazovaya, Platon Eliseev, Elena Miteneva, Dmitrii Polev, Igor Mokrousov, and Andrei Maryandyshev

Corresponding Author(s): Anna Vyazovaya, Sankt-Peterburgskij naucno-issledovatel'skij institut epidemiologii i mikrobiologii imeni Pastera

Review Timeline:

Submission Date:	October 21, 2025
Editorial Decision:	January 13, 2026
Revision Received:	January 23, 2026
Accepted:	February 5, 2026

Editor: Sophia Georghiou

Reviewer(s): Disclosure of reviewer identity is with reference to reviewer comments included in decision letter(s). The following individuals involved in review of your submission have agreed to reveal their identity: Imane Chaoui (Reviewer #1)

Transaction Report:

DOI: <https://doi.org/10.1128/spectrum.03431-25>

Re: Spectrum03431-25 (**Key impact of Beijing strains including new resistant clusters on spread of multidrug-resistant tuberculosis in northern Russia**)

Dear Dr. Anna Vyazovaya:

Thank you for the privilege of reviewing your work. Below you will find my comments, instructions from the Spectrum editorial office, and the reviewer comments.

Revision Guidelines

Sincerely,
Sophia Georghiou
Editor
Microbiology Spectrum

Reviewer #1 (Comments for the Author):

This paper presents a clear, well-structured, and scientifically rigorous investigation that effectively integrates molecular epidemiology with public health relevance in a region of Northern Russia characterized by high TB and MDR-TB rates. The study demonstrates that the Beijing genotype of *Mycobacterium tuberculosis* continues to dominate in Northern Russia, with particularly high levels of drug resistance concentrated in several key sublineages. The identification of long-term transmission patterns and drug-resistant clusters provides valuable insights that can directly inform regional TB control strategies.

Major Concerns

The authors note that the recent COVID-19 pandemic has influenced tuberculosis epidemiology and the population structure of *M. tuberculosis*. Given this, the manuscript would be strengthened by including a comparison of the pre- and post-pandemic periods. Such an analysis is important for assessing current transmission dynamics and the potential emergence of new clones.

Minor Comments

Sentence corrections:

"The WHO's 2021 catalogue of *M. tuberculosis* mutations provides for enhanced genotypic resistance 105 detection."
→ The WHO's 2021 catalogue of *M. tuberculosis* mutations provides enhanced genotypic resistance detection.

"The Arkhangelsk region is the largest by area province of northern European Russia."
→ The Arkhangelsk region is the largest province by area in northern European Russia.

The sentence "The global burden of rifampicin-resistant TB (RR-TB) rose to 450 000 98 new cases in 2021, according to the World Health Organization (WHO) (1)." should be checked and corrected for clarity and numerical formatting.

The authors may consider incorporating more recent epidemiological estimates from the Global Tuberculosis Report 2025, if available.

The term "medically important MDR clusters" should be defined more clearly. What criteria were used to classify clusters as medically important (e.g., size, transmission characteristics, clinical outcomes, resistance profiles)?

The following sentence requires clarification:

"Both the BACTEC MGIT 960 system and the solid Löwenstein-Jensen method are employed to ensure rapid and accurate drug susceptibility testing, with MGIT providing timely results for clinical decision-making and LJ serving as a confirmatory phenotypic gold standard in accordance with national guidelines. Both methods are used in this setting in routine ?"

The authors should revise this section to clearly state whether both methods are routinely used in the study setting and how they complement each other operationally.

Reviewer #2 (Comments for the Author):

In this study a total of 184 patients with pulmonary tuberculosis were identified in the Arkhangelsk region of Russia from January 1 to December 31, 2018. For 90 of them, *M. tuberculosis* isolates were recovered. After some exclusions, the authors report the genotyping data for 88 *M. tuberculosis* isolates from 88 patients.

General comments:

overall the results of the manuscript are interesting and the information supplied is important for continued interpretation of infection cycles and transmission routes in tuberculosis research. The authors also describe well the problem of multidrug resistance and the resistance profiles of the panel of strains studied. The manuscript is clearly written and I only have few suggestions for improvement.

Specific comments:

Line 229 - 232: here 51 sensitive, and 30 MDR *Mtb* samples are described, representing 81 of the 88 strains studied. It might be helpful to mention to what correspond the remaining 7 isolates . Are these mono resistant strains ?

Line 280 -283: please define how many strains were in each cluster from which the DNA samples were genome sequenced, including the total number of sequenced isolates. According to Fig 2, the total might be 6+4 strains sequenced, but it would be more convenient for the reader to find this information also in the text.

Line 411. In a recent publication by Bonnet et al., *Microbiology Spectrum*, 2025, some specific mutations in regulatory genes in *Mtb* strains from the B0/W148 were investigated, suggesting that these highly circulating MDR clones might not have an enhanced secretion of virulence factors, such as ESAT-6. It might be of interest if the authors could check in their sequences if their strains from the B0/W148 cluster from the Arkhangelsk region carried the same mutations in these regulatory genes as the strains described in this recent publication from France. It might well be that high spreading potential of selected epidemic strains might not be linked to increased virulence factor expression or secretion, as also suggested by a recent paper by Culviner et al., *Cell* 2025. This issue might be worth being discussed in the manuscript.

Re: Spectrum03431-25 (Key impact of Beijing strains including new resistant clusters on spread of multidrug-resistant tuberculosis in northern Russia)

Response to Reviewers

REVIEWER #1:

This paper presents a clear, well-structured, and scientifically rigorous investigation that effectively integrates molecular epidemiology with public health relevance in a region of Northern Russia characterized by high TB and MDR-TB rates. The study demonstrates that the Beijing genotype of *Mycobacterium tuberculosis* continues to dominate in Northern Russia, with particularly high levels of drug resistance concentrated in several key sublineages. The identification of long-term transmission patterns and drug-resistant clusters provides valuable insights that can directly inform regional TB control strategies.

Major Concerns

The authors note that the recent COVID-19 pandemic has influenced tuberculosis epidemiology and the population structure of *M. tuberculosis*. Given this, the manuscript would be strengthened by including a comparison of the pre- and post-pandemic periods. Such an analysis is important for assessing current transmission dynamics and the potential emergence of new clones.

ANSWER: We agree and we plan such new study that will likely take up to 2 years.

In the present study, we had the objective to compare the pre-pandemic data just before the pandemic (“old normal”) with those published in the early 2000s. In the continuation study, we will include both *M. tuberculosis* isolates collected from two time-periods: during the pandemic, in 2020-2021, and 5 years after it was declared ended (in 2023).

The text was revised (Conclusions, Page 20, last para).

Minor Comments

Sentence corrections:

"The WHO's 2021 catalogue of *M. tuberculosis* mutations provides for enhanced genotypic resistance detection."

→ The WHO's 2021 catalogue of *M. tuberculosis* mutations provides enhanced genotypic resistance detection.

ANSWER: Corrected (Page 5, lines 108-110). Please note that all changes are highlighted in the separate file “manuscript with marked up changes”

"The Arkhangelsk region is the largest by area province of northern European Russia."

→ The Arkhangelsk region is the largest province by area in northern European Russia.

ANSWER: Corrected (Page 6, last para).

The sentence "The global burden of rifampicin-resistant TB (RR-TB) rose to 450 000 98 new cases in 2021, according to the World Health Organization (WHO) (1)." should be checked and corrected for clarity and numerical formatting.

The authors may consider incorporating more recent epidemiological estimates from the Global Tuberculosis Report 2025, if available.

ANSWER: We updated this information and replaced Ref #1 with the most recent Global Tuberculosis Report 2025. The text was also revised (Page 5, 1st para).

The term "medically important MDR clusters" should be defined more clearly. What criteria were used to classify clusters as medically important (e.g., size, transmission characteristics, clinical outcomes, resistance profiles)?

ANSWER: There is no a single consensus definition of such clusters and not all are necessarily MDR. Some clusters are considered medically important because of their clinical and/or epidemiological significance i.e. association with more severe course of disease, increased virulence (e.g., assessed in the animal model), or increased transmissibility (e.g., assessed by high clustering rate or in vitro growth rate).

Indeed, the reasons of success may vary for different clusters/genotypes: some drug susceptible strains may be important because of the increased transmissibility and increased virulence, e.g. LAM SIT264 strain [Mokrousov et al., 2023], Beijing susceptible subtype in South Africa [...]. On the other hand, there are hypervirulent but strictly endemic strains with a limited spread (e.g. Beijing 14717-15 "Buryat" genotype [Vinogradova et al., 2021]). Certain genotypes have an unusual manifestation of the virulence characteristics, e.g. mainly susceptible Haarlem genotype as demonstrated in mouse and macrophage models [Reiling et al., 2013]. Perhaps a certain balance between drug resistance and virulence may result in a wide spread of Beijing Central Asian/Russian subtype (~94-32-cluster) over all Russia [Vyazovaya et al., 2023].

This is described in more detail in the revised text and we added relevant references (Page 6, 2nd para).

The following sentence requires clarification:

"Both the BACTEC MGIT 960 system and the solid Löwenstein-Jensen method are employed to ensure rapid and accurate drug susceptibility testing, with MGIT providing timely results for clinical decision-making and LJ serving as a confirmatory phenotypic gold standard in accordance with national guidelines. Both methods are used in this setting in routine ?"

The authors should revise this section to clearly state whether both methods are routinely used in the study setting and how they complement each other operationally.

ANSWER: This information on DST was rechecked and corrected. Solid LJ medium was used only for culture, while Bactec MGIT was the only method used for DST (Page 8).

Reviewer #2:

In this study a total of 184 patients with pulmonary tuberculosis were identified in the Arkhangelsk region of Russia from January 1 to December 31, 2018. For 90 of them, M. tuberculosis isolates were recovered. After some exclusions, the authors report the genotyping data for 88 M. tuberculosis isolates from 88 patients.

General comments:

overall the results of the manuscript are interesting and the information supplied is important for continued interpretation of infection cycles and transmission routes in tuberculosis research. The authors also describe well the problem of multidrug resistance and the resistance profiles of the panel of strains studied. The manuscript is clearly written and I only have few suggestions for improvement.

Specific comments:

Line 229 - 232: here 51 sensitive, and 30 MDR Mtb samples are described, representing 81 of the 88 strains studied. It might be helpful to mention to what correspond the remaining 7 isolates. Are these mono resistant strains ?

ANSWER: Yes, the remaining 7 isolates were: 4 INH-monoresistant and 3 were resistant to two drugs (2 – INH and EMB, and 1 resistant to INH and OFL). This information was added in the revised version (Page 11, 1st para).

Line 280 -283: please define how many strains were in each cluster from which the DNA samples were genome sequenced, including the total number of sequenced isolates. According to Fig 2, the total might be 6+4 strains sequenced, but it would be more convenient for the reader to find this information also in the text.

ANSWER: Cluster 3828-32 included 7 isolates and all were submitted to WGS but one isolate failed in WGS due to poor DNA quality or low concentration. In its turn, all 4 isolates of cluster 10167-32 were submitted to WGS and the high-quality data were obtained for all of them. We edited the text (Page 13, 1st and 2nd para).

Line 411. In a recent publication by Bonnet et al., Microbiology Spectrum, 2025, some specific mutations in regulatory genes in Mtb strains from the B0/W148 were investigated, suggesting that these highly circulating MDR clones might not have an enhanced secretion of virulence factors, such as ESAT-6. It might be of interest if the authors could check in their sequences if their strains from the B0/W148 cluster from the Arkhangelsk region carried the same mutations in these regulatory genes as the strains described in this recent publication from France. It might well be that high spreading potential of selected epidemic strains might not be linked to increased virulence factor expression or secretion, as also suggested by a recent paper by Culviner et al., Cell 2025. This issue might be worth being discussed in the manuscript.

ANSWER: WhiB6 T51P and KdpDE del_TG mutations were previously proposed as specific for Beijing B0/W148 based on analysis of large and diverse collection.

In our study, VNTR type 3828-32 is clustering on the dendrogram with type 100-32, the major variant of the Russian B0/W148 strain, so they belong to Beijing B0/W148 subtype. We checked WGS data of the 3828-32 strains and as expected all of them have these particular mutations in transcriptional regulators WhiB6 and KdpDE. In this sense, they do not differ from other Beijing B0/W148 strains.

We added some relevant text and references in the revised version (Page 17, 2nd para).

Re: Spectrum03431-25R1 (**Key impact of Beijing strains including new resistant clusters on spread of multidrug-resistant tuberculosis in northern Russia**)

Dear Dr. Anna Vyazovaya:

Your manuscript has been accepted, and I am forwarding it to the ASM production staff for publication. Your paper will first be checked to make sure all elements meet the technical requirements. ASM staff will contact you if anything needs to be revised before copyediting and production can begin. Otherwise, you will be notified when your proofs are ready to be viewed.

Sincerely,
Sophia Georghiou
Editor
Microbiology Spectrum

Reviewer #1 (Comments for the Author):

none

Reviewer #2 (Comments for the Author):

The authors have responded in a concise and appropriate fashion to the comments and requests of the reviewers. and have thereby improved the manuscript.